META-RESEARCH

# A retrospective analysis of the peer review of more than 75,000 Marie Curie proposals between 2007 and 2018

**Abstract** Most funding agencies rely on peer review to evaluate grant applications and proposals, but research into the use of this process by funding agencies has been limited. Here we explore if two changes to the organization of peer review for proposals submitted to various funding actions by the European Union has an influence on the outcome of the peer review process. Based on an analysis of more than 75,000 applications to three actions of the Marie Curie programme over a period of 12 years, we find that the changes – a reduction in the number of evaluation criteria used by reviewers and a move from in-person to virtual meetings – had little impact on the outcome of the peer review process. Our results indicate that other factors, such as the type of grant or area of research, have a larger impact on the outcome.

**DAVID G PINA[†*], IVAN BULJAN[†], DARKO HREN AND ANA MARUŠIĆ**

## Introduction

Peer review is widely used by journals to evaluate research papers (*Bornmann, 2011*; *Bornmann et al., 2010*; *Jackson et al., 2011*; *Baethge et al., 2013*), and by funding agencies to evaluate grant applications (*Cicchetti, 1991*; *Wessely, 1998*; *Reinhart, 2009*; *Guthrie et al., 2018*). However, research into the use of peer review to assess grant applications has been hampered by the unavailability of data and the range of different approaches to peer review adopted by funding agencies. As such, the majority of studies have relied on relatively small samples (*Cole et al., 1981*; *Fogelholm et al., 2012*; *Hodgson, 1997*; *Mayo et al., 2006*; *Pier et al., 2018*), although some studies have been performed on larger samples (see, for example, *Mutz et al., 2012*). To date most studies acknowledge the need for improvements (*Demicheli et al., 2007*; *Gallo et al., 2014*; *Graves et al., 2011*; *Jirschitzka et al., 2017*; *Marsh et al., 2008*; *Sattler et al., 2015*; *Shepherd et al., 2018*; *Bendiscioli, 2019*): in particular, it has been shown that the peer

review of grant applications is subject to various forms of bias (*Lee et al., 2013*; *Witteman et al., 2019*).

The peer review process at many funding agencies concludes with an in-person meeting at which the reviewers discuss and compare the applications they have reviewed. However, some funding agencies are replacing these meetings with virtual ones to reduce both their cost and carbon footprint, and to ease the burdens placed on reviewers. A number of small-scale studies have shown that moving from in-person to virtual meetings had little impact on the outcome of the peer-review process (*Gallo et al., 2013*; *Carpenter et al., 2015*; *Obrecht et al., 2007*), and a large-scale study by some of the present authors involving almost 25,000 grant applications to the European Union's Seventh Framework Programme (FP7) reported (along with other findings) that evaluating research proposals remotely would be, to a certain extent, feasible and reliable (*Pina et al., 2015*).

**\*For correspondence:**
David.Pina@ec.europa.eu

[†]These authors contributed equally to this work

**Competing interests:** The authors declare that no competing interests exist.

Here we explore if two changes in the way peer review was used to evaluate proposals to a number of European Union (EU) funding programmes had any impact on the outcome of the peer review process. The first change came in 2014, when FP7 gave way to the Horizon 2020 (H2020) programme: one consequence of this was that the number of evaluation criteria applied to assess applications was reduced from four or more to three: excellence, impact, and implementation. The second change was the replacement of in-person meetings by virtual meetings for a number of funding actions.

Ensuring that the evaluation process remained stable and reliable during these changes was a priority for the EU. To assess the impact of these two changes we analyzed almost 25,000 proposals to FP7 and more than 50,000 proposals to H2020 over a period of 12 years.

## Results

The European Union has been funding researchers and projects under actions named after Marie Curie since 1996. Marie Curie Actions (MCA) were part of FP7, which ran from 2007 to 2013, and were renamed Marie Skłodowska-Curie Actions (MSCA) when H2020 started in 2014. MCA had a budget of €4.7 billion, which increased to €6.2 billion for MCSA. The MCSA programme awards funding to several actions, namely the Individual Fellowships (partial successor of the Intra-European Fellowships in FP7), Innovative Training Networks (called Initial Training Networks in FP7), and Research and Innovation Staff Exchange (partial successor of the Industry-Academia Pathways and Partnerships in FP7). In terms of number of applications, Individual Fellowships (IF) is the largest action, receiving more than 43,000 applications between 2014 to 2018. The success rate for applications varies from below 10% for Innovative Training Networks (ITN) to about 15% for Individual Fellowships and around 20–30% for Research and Innovation Staff Exchange (RISE), depending on the year. Calls for proposals are organized on a yearly basis, and the number of proposals evaluated each year make MCA/MSCA well suited as a system for studying the peer review of grant applications and proposals.

The MCA/MSCA evaluation process has been explained elsewhere (*Pina et al., 2015*), and consists of two steps. The first step, the individual evaluation, is done entirely remotely: each proposal is assessed by (typically) three reviewers, with each reviewer producing an Individual Evaluation Report (IER), and scoring each criterion on a scale from 0 (fail) to 5 (excellent), with a single decimal resolution. During this step, the three reviewers are unaware of each other's identity.

**Table 1.** Number of proposals, number of evaluation criteria, and format of the consensus phase for the three different Marie Curie actions between 2007 and 2018.

| Grant call* | | 2007 | 2008 | 2009 | 2010 | 2011 | 2012 | 2013 | 2014 | 2015 | 2016 | 2017 | 2018 |
|---|---|---|---|---|---|---|---|---|---|---|---|---|---|
| IEF/IF | No. proposals | 1686 | 1753 | 2392 | 2835 | 3302 | 3708 | 4917 | 7397 | 8364 | 8805 | 8940 | 9658 |
| | No. evaluation criteria | 5 | 5 | 5 | 5 | 5 | 5 | 5 | 3 | 3 | 3 | 3 | 3 |
| | Consensus format | on-site | on-site | on-site | on-site | on-site | on-site | on-site | on-site | on-site | remote | remote | remote |
| ITN | No. proposals | † | 886 | † | 858 | 909 | 892 | † | 1149 | 1558 | † | 1702 | 1634 |
| | No. evaluation criteria | | 4 | | 4 | 4 | 4 | | 3 | 3 | | 3 | 3 |
| | Consensus format | | on-site | | on-site | on-site | on-site | | on-site | on-site | | remote | remote |
| IAPP/RISE | No. proposals | 102 | 141 | 356 | † | 160 | † | † | 200 | 361 | 366 | 321 | 272 |
| | No. evaluation criteria | 4 | 4 | 4 | | 4 | | | 3 | 3 | 3 | 3 | 3 |
| | Consensus format | on-site | on-site | on-site | | on-site | | | on-site | on-site | on-site | on-site | on-site |

*The Results section describes how the three actions within the EU's Marie Curie funding programme (IEF/IF, ITN and IAPP/RISE) changed between 2007 and 2018.

† Data for these calls were not considered for the following reasons. ITN 2007: organized as a two-stage evaluation process; ITN 2008 and IAPP 2010: no calls organized for these years; ITN 2013, IAPP 2012 and IAPP 2013: data not accessible for technical reasons; ITN 2016: organized with four reviewers per proposal instead of three.

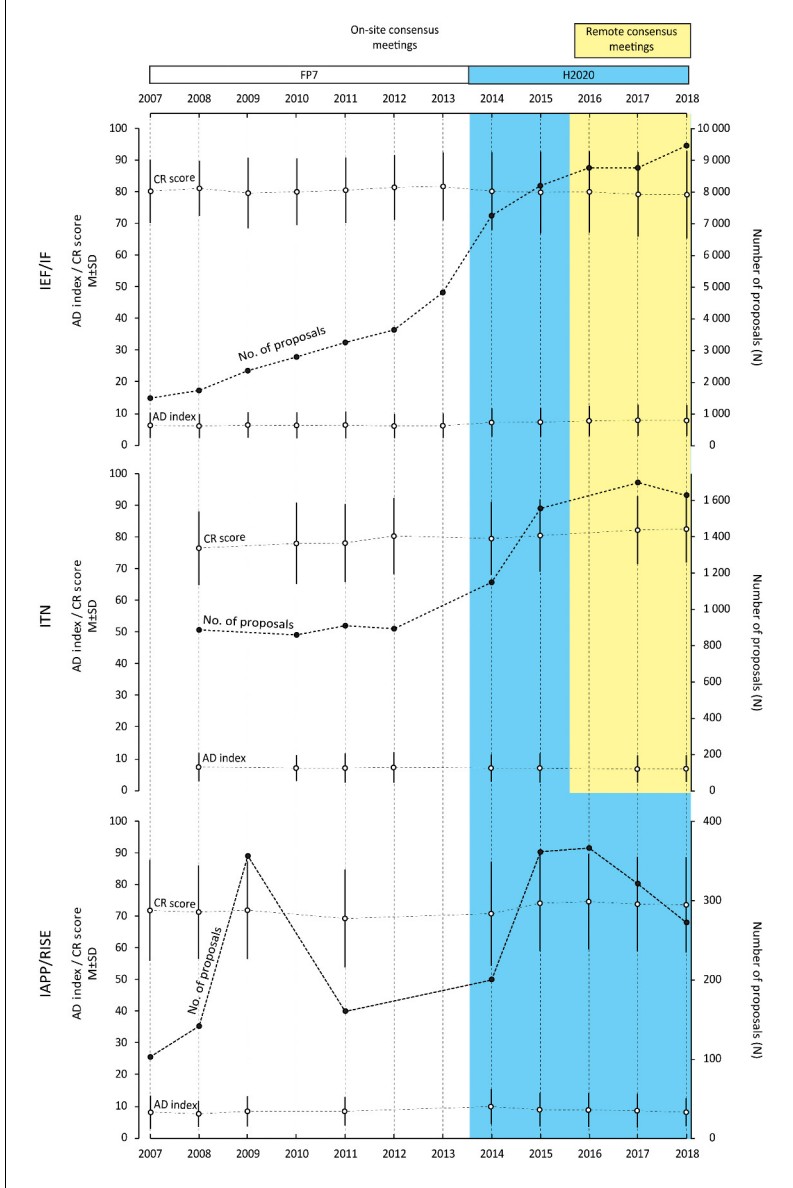

**Figure 1.** Number of proposals, CR scores (mean and SD) and AD indices (mean and SD) for the three different Marie Curie actions between 2007 and 2018. We studied 75,624 proposals evaluated under the EU's Marie Curie funding programme between 2007 and 2018 to investigate if two changes to the way peer review is organized – a reduction in the number of evaluation criteria in 2014, and a move from in-person to remote consensus meetings in 2016 – influenced the outcome of the peer review process. In the white region (which corresponds to FP7) four or more criteria were used to evaluate proposals and consensus meetings were in-person. In the coloured region (which correspond to H2020) three criteria were used to evaluate proposals: consensus meetings remained in-person in the blue region, but became remote/virtual in the yellow region. The Results section describes how the three Marie Curie actions (IEF/IF, ITN and IAPP/RISE) changed over this period. Data for certain calls were not considered for the following reasons. ITN 2007: organized as a two-stage evaluation process; ITN 2008 and IAPP 2010: no calls organized for these years; ITN 2013, IAPP 2012 and IAPP 2013: data not accessible for technical reasons; ITN 2016: organized with four reviewers per proposal.

Once the IER are completed, a consensus meeting is organized for each proposal, with the reviewers agreeing on a consolidated set of comments and scores that are summarized in a Consensus Report (CR). Although based on the initial IER scores, the final CR score is usually not an average of these scores. The CR is corrected for typos and other clerical errors to produce an Evaluation Summary Report (ESR): however, in practice, this has the same content and score as the CR, so we will refer to the CR score throughout this article. Ranked lists of proposals are established based on their CR scores, determining the priority order for funding, and the top-scored proposals get funded up to the available call budget.

Under H2020, all MSCA proposals are scored (for both IER and CR) on three evaluation criteria: excellence (which accounts for 50% of the score), impact (30%), and implementation (20%). Under FP7, the number of evaluation criteria varied with the type of action, as did the weighting attached to each: there were five criteria for Intra-European Fellowships (IEF) and four for Initial Training Networks (ITN) and Industry-Academia Pathways and Partnerships (IAPP). Under both FP7 and H2020 the IER and CR scores are a number between 0 and 100.

In our analysis, for each proposal we used the average deviation (AD) index as a measure of the (dis)agreement between the reviewers (*Burke et al., 1999*; *Burke and Dunlap, 2002*). This index is the sum of the absolute differences between each individual reviewer's score (IER) and the average (mean) score for a given proposal (AVIER), divided by the number of reviewers. For a proposal evaluated by three reviewers with scores IER1, IER2 and IER3, then the AD index is ($|IER1 - AVIER| + |IER2 - AVIER| + |IER3 - AVIER|$)/3. The AD index does not require the specification of null distribution and returns value in the units of the original scale (0–100 in our case), making its interpretation easier and more pragmatic (*Smith-Crowe et al., 2013*): the closer the AD index is to zero, the greater the agreement between reviewers. We also calculated the difference between the CR score and the AVIER (CR-AVIER) for each proposal.

Categorical data are presented as aggregated sums, frequencies and percentages, and continuous data as means and standard deviations. The differences between agreement groups were tested with one-way ANOVA. We assessed the associations between the CR scores and the average of IER scores, and between the

CR scores and AD indices using Pearson's correlation coefficient. We used the interrupted time series analysis to assess how changes in the organization of peer review influenced CR scores and AD indices over the years (*Cochrane Effective Practice and Organisation of Care (EPOC), 2017*).

*Table 1* shows the number of evaluated proposals to each of the three actions (IEF/IF, ITN, IAPP/RISE) between 2007 and 2018, along with the number of evaluation criteria and the format of the consensus meeting (ie, on-site or remote). For each of the three actions *Figure 1* plots the number of proposals, the mean CR scores and the mean AD indices over the same period. The two changes to the organization of peer review made during this period appear to have had very little impact on the mean CR scores or the mean AD indices, with the changes due to the reduction in the number of evaluation criteria being more pronounced than those due to the move from in-person to remote or virtual meetings (*Table 2*). The changes observed were very small and non-systematic, implying that it may probably be attributable to the large number of analyzed proposals and relatively few observation points, rather than some meaningful trend.

*Table 3* shows the number of evaluated proposals, the mean CR scores and the mean AD indices for each of the three actions over three time periods (2007–2013; 2014–2015; 2016–2018), broken down by scientific panels (see Methods). The two panels with the most proposals for Individual Fellowships during the whole 2007–2018 period were life sciences and economics and social sciences and humanities; whereas for ITN the panels with the most applications were life sciences and engineering, and for IAPP/RISE there was a predominance of engineering proposals. The mean CR scores and AD indices remained stable over the period studied.

We also studied the difference between the CR scores and the average of the IER scores, and the distribution of this difference is plotted in *Figure 2* (along with the distributions for the AD indices and CR scores) for the three actions over different time periods. The distribution for the difference in scores is bell-shaped, with a maximum at zero difference; moreover, we found that the absolute value of this difference was two points or less (on a scale of 0–100) for 37,900 proposals (50.1% of the total), and 10 points or less for 72,527 proposals (95.9%). We also found (using Pearson's correlation coefficients) that the CR scores and the average of the IER scores were highly correlated for all three types of grants (Table A1 in *Supplementary file 2*). Higher CR scores also tended to have lower AD indices for all three actions (*Figure 3*), with

**Table 2.** Results of interrupted time series analyses for mean CR scores and mean AD indices for the three different Marie Curie actions between 2007 and 2018.

| | | | Pre-intervention slope coefficient (95% CI)* | Post-intervention slope coefficient (95% CI)* | Change in slope (95% CI)* |
|---|---|---|---|---|---|
| Change in the number of evaluation criteria† | AD index | IEF/IF | -0.01 (-0.06 to 0.04) | 0.19 (0.09 to 0.29) | 0.20 (0.09 to 0.31) |
| | | ITN | -0.03 (-0.18 to 0.24) | -0.09 (-0.30 to 0.12) | -0.06 (-0.24 to 0.36) |
| | | IAPP/RISE | 0.11 (-0.26 to 0.48) | -0.39 (-0.65 to -0.13) | -0.50 (-0.95 to -0.05) |
| | CR Score | IEF/IF | 0.22 (-0.05 to 0.49) | -0.29 (-0.73 to 0.16) | -0.51 (-1.02 to 0.01) |
| | | ITN | 1.15 (0.41 to 1.89) | 1.07 (0.32 to 1.81) | -0.08 (-1.13 to 0.96) |
| | | IAPP/RISE | -0.69 (-2.06 to 0.68) | 0.53 (-0.44 to 1.50) | 1.22 (-0.44 to 2.88) |
| Change in the consensus format† | AD index | IEF/IF | 0.12 (-0.02 to 0.26) | 0.09 (-0.47 to 0.65) | -0.02 (-0.56 to 0.60) |
| | | ITN | -0.04 (-0.13 to -0.06) | 0.10 (-0.47 to 0.67) | 0.14 (-0.44 to 0.72) |
| | CR Score | IEF/IF | 0.02 (-0.03 to 0.03) | -0.49 (-1.76 to 0.78) | -0.51 (-1.81 to 0.79) |
| | | ITN | 0.76 (0.24 to 1.27) | 0.30 (-2.7 to 3.39) | -0.46 (-3.59 to 2.67) |

\* Positive (negative) values of the slope coefficient can be interpreted as an increase (decrease) in the average points per call (on a scale of 1–100).

† The change in the number of evaluation criteria occurred in 2014 with the transition from FP7 to H2020; the change in the consensus format occurred in 2016 for IF and ITN. The Results section describes how the three actions within the EU's Marie Curie funding programme (IEF/IF, ITN and IAPP/RISE) changed between 2007 and 2018.

AD index: average deviation index; CI: confidence interval; CR: Consensus Report.

**Table 3.** Number of proposals, CR scores (mean and SD) and AD indices (mean and SD), broken down by scientific panel, for the three different Marie Curie actions for three time periods between 2007 and 2018.

| | | No. proposals (% total) | | | Mean CR score (SD) | | | Mean AD index (SD) | | |
|---|---|---|---|---|---|---|---|---|---|---|
| | | 2007–13 | 2014–15 | 2016–18 | 2007–13 | 2014–15 | 2016–18 | 2007–13 | 2014–15 | 2016–18 |
| IEF/IF | Overall | 20,593 | 15,761 | 27,403 | 80.4 (10.4) | 79.6 (12.6) | 79.0 (13.3) | 6.1 (4.0) | 7.1 (4.6) | 7.6 (4.9) |
| | CHE | 2204 (10.7) | 1837 (11.7) | 3449 (12.6) | 81.1 (9.2) | 79.8 (11.3) | 79.9 (12.6) | 5.6 (3.6) | 6.5 (4.1) | 6.9 (4.4) |
| | ECOSOC | 4228 (20.5) | 3473 (22.0) | 6614 (24.1) | 78.7 (12.4) | 78.2 (14.3) | 76.5 (15.0) | 7.6 (4.7) | 8.3 (5.1) | 9.0 (5.5) |
| | ENG | 1888 (9.2) | 1935 (12.3) | 3249 (11.9) | 78.0 (11.2) | 76.6 (13.7) | 77.4 (14.5) | 6.4 (4.1) | 7.8 (4.7) | 7.9 (5.1) |
| | ENV | 2731 (13.3) | 2058 (13.1) | 3531 (12.9) | 81.1 (9.7) | 79.7 (12.7) | 79.7 (12.9) | 5.7 (3.7) | 6.8 (4.4) | 7.5 (4.6) |
| | LIF | 6408 (31.1) | 4304 (27.3) | 7065 (25.8) | 81.5 (9.5) | 81.3 (11.2) | 80.9 (12.1) | 5.5 (3.5) | 6.5 (4.2) | 7.2 (4.5) |
| | MAT | 665 (3.2) | 369 (2.3) | 611 (2.2) | 78.4 (10.1) | 79.0 (13.3) | 79.4 (12.9) | 6.2 (4.1) | 7.5 (4.8) | 7.1 (4.5) |
| | PHY | 2469 (12.0) | 1785 (11.3) | 2884 (10.5) | 81.0 (9.0) | 81.4 (10.4) | 80.1 (11.1) | 5.4 (3.5) | 5.9 (4.0) | 6.2 (4.0) |
| | | 2007–13 | 2014–15 | 2017–18 | 2007–13 | 2014–15 | 2017–18 | 2007–13 | 2014–15 | 2017–18 |
| ITN | Overall | 3545 | 2707 | 3336 | 78.0 (12.2) | 79.9 (11.5) | 82.1 (10.6) | 7.2 (4.5) | 7.1 (4.5) | 6.8 (4.3) |
| | CHE | 398 (11.2) | 316 (11.7) | 397 (11.9) | 79.4 (10.4) | 81.6 (9.9) | 84.7 (8.7) | 7.0 (4.3) | 6.2 (4.3) | 6.2 (3.7) |
| | ECOSOC | 381 (10.7) | 245 (9.1) | 363 (10.9) | 73.6 (15.1) | 78.0 (13.5) | 81.1 (12.3) | 8.7 (5.3) | 8.7 (5.5) | 7.8 (4.8) |
| | ENG | 799 (22.5) | 794 (29.3) | 1039 (31.1) | 76.4 (12.3) | 78.3 (12.3) | 80.5 (10.5) | 7.5 (4.4) | 7.3 (4.3) | 7.0 (4.1) |
| | ENV | 428 (12.1) | 335 (12.4) | 404 (12.1) | 78.1 (11.3) | 80.0 (11.0) | 83.2 (10.9) | 6.7 (4.3) | 6.6 (4.0) | 6.6 (4.2) |
| | LIF | 1047 (29.5) | 764 (28.2) | 868 (26.0) | 79.3 (12.5) | 80.7 (11.0) | 82.6 (10.6) | 7.0 (4.4) | 7.1 (4.3) | 6.9 (4.5) |
| | MAT | 60 (1.7) | 42 (1.6) | 44 (1.3) | 77.3 (8.2) | 77.2 (9.6) | 79.7 (10.6) | 8.1 (4.3) | 8.0 (5.5) | 6.5 (4.6) |
| | PHY | 432 (12.2) | 211 (7.8) | 221 (6.6) | 80.6 (10.2) | 82.3 (9.5) | 83.8 (9.1) | 6.5 (4.3) | 6.2 (4.0) | 5.9 (3.8) |
| | | 2007–13 | 2014–18 | | 2007–13 | 2014–18 | | 2007–13 | 2014–18 | |
| IAPP/RISE | Overall | 759 | 1520 | | 70.9 (15.3) | 73.3 (15.2) | | 8.1 (4.7) | 8.7 (5.2) | |
| | CHE | 63 (8.3) | 142 (9.3) | | 69.4 (15.3) | 77.6 (13.5) | | 7.6 (4.6) | 7.2 (4.5) | |
| | ECOSOC | 68 (9.0) | 283 (18.6) | | 67.1 (18.1) | 71.3 (16.1) | | 8.4 (5.2) | 9.4 (5.5) | |
| | ENG | 296 (39.0) | 462 (30.4) | | 71.3 (13.6) | 72.0 (15.8) | | 7.8 (4.5) | 8.9 (5.1) | |
| | ENV | 84 (11.1) | 204 (13.4) | | 72.1 (16.3) | 74.5 (15.4) | | 8.4 (4.5) | 8.9 (5.5) | |
| | LIF | 203 (26.7) | 231 (15.2) | | 63.9 (16.3) | 72.4 (14.5) | | 8.5 (4.6) | 9.1 (5.2) | |
| | MAT | 6 (0.8) | 47 (3.1) | | 63.9 (24.6) | 73.2 (14.4) | | 9.2 (4.6) | 8.1 (5.3) | |
| | PHY | 39 (5.1) | 151 (9.9) | | 75.0 (11.5) | 77.0 (16.1) | | 8.2 (5.6) | 7.7 (4.6) | |

The Results section describes how the three actions within the EU's Marie Curie funding programme (IEF/IF, ITN and IAPP/RISE) changed between 2007 and 2018.

CHE: Chemistry; ECOSOC: Economics and social sciences and humanities; ENG: Engineering; ENV: Environment; LIF: Life sciences; MAT: Mathematics; PHY: Physics.

some panels (economics and social sciences and humanities) and some actions (IAPP/RISE) having higher mean AD indices than other panels and actions (*Table 3*). Overall, for all proposals included in the analysis, the mean value of the AD indices was 7.02 (SD = 4.56), with 59,500 proposals (78.7% of the total) having an AD index of 10 or less (on a scale of 0–100). This suggests a high level of agreement between the reviewers.

To explore if there was a relationship between the level of agreement (or disagreement) among the reviewers and the CR scores,

we divided the H2020 proposals into three groups and calculated the mean CR scores for each group. In the 'full agreement' group all the three absolute differences between IER scores of each pair of reviewers were 10 points or less. In the 'no agreement' group all the absolute differences were above 10 points. In the 'other' group at least one absolute difference was 10 points or less, and at least one was more than 10 points. Of the 50,727 proposals we studied, most (31,803; 62.7% of the total) were in the 'other' group, followed by the 'full agreement' group (12,840; 25.3%), and the 'no agreement'

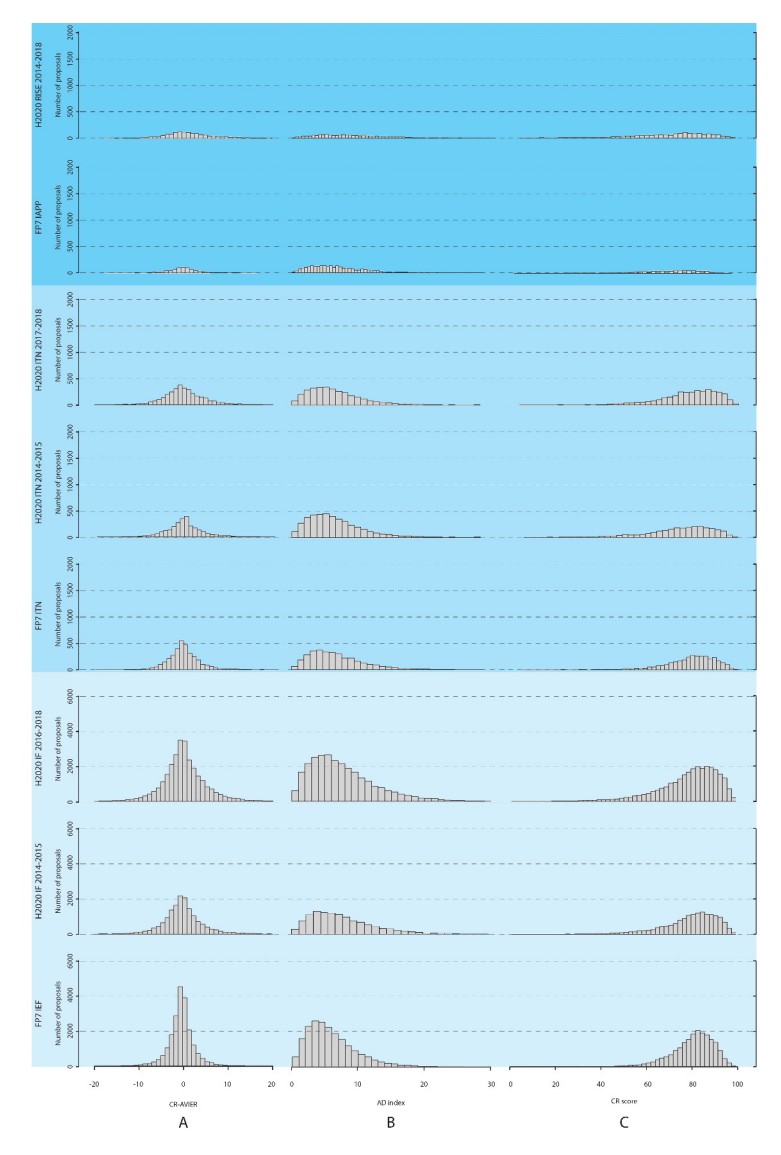

**Figure 2.** Distributions of CR scores, AD indices, and the difference between the CR scores and the average of for IER scores, for the three different Marie Curie actions between 2007 and 2018. Proposals are evaluated by three reviewers to produce Individual Evaluation Reports (IER), which are then consolidated to produce a Consensus Report (CR); this process also involves agreeing a CR score, which does not have to be the average of the IER scores. (**A**) The left column shows the distribution of the difference between the CR scores and the average of the IER scores; (**B**) the middle column shows the distribution of the AD (average deviation) indices; and (**C**) the left column shows the distribution of the CR scores. The distributions are shown for the three Marie Curie actions (IEF/IF, ITN and IAPP/RISE) during different time periods. The Results section describes how these actions changed over this period.

group (6,084; 12.0%). In all cases, the 'full agreement' group had the highest mean CR scores, followed by the 'other' group and the 'no agreement' group (*Table 4*). For the IF and ITN actions the 'full agreement' group was generally bigger than the 'no agreement' group by a

factor of about two; for RISE the two groups tended to be comparable in size (with the exception of 2014, when the 'no agreement' group was much larger). We also looked at these three groups by scientific panel and found no deviations from the general trends observed at the action level (*Table 5*). Across all H2020 proposals, those in the 'full agreement' group had an average CR score of 85.1 (SD = 10.8), whereas those in the 'no agreement' group had a CR score of 70.3 (SD = 13.3).

We also identified 3,097 H2020 proposals for which the difference between the CR scores and the average of the IER scores was greater than 10 points (Table A2 in *Supplementary file 2*): in 38.9% of cases the difference was positive (meaning that the CR score was higher than the average of the IER scores), and in 61.1% of cases was negative (meaning that the CR score was lower than the average of the IER scores). The mean CR score for this subsample (67.8, SD = 18.37) was lower than that for all (FP7 and H2020) proposals (79.5, SD = 12.4), and the mean AD index (12.86, SD = 6.33) was higher than that for H2020 proposals (7.38, SD = 4.74).

That result indicates that proposals having a greater discrepancy between CR scores and the average of the IER scores show higher AD indices, and end up being more difficult to reach consensus. Another clear finding of our study is that the more reviewers disagree about a proposal, the lower the proposal's final score. This trend was observed consistently over the years, for all type of actions, and in all scientific fields, confirming the observations from the FP7 dataset (*Pina et al., 2015*).

## Discussion

Our analysis of over 75,000 thousand proposals from both FP7 and H2020, covering the period from 2007 to 2018, suggests that the peer review process used to evaluate these proposals is resistant to organizational changes, such as the reduction of the number of evaluation criteria and the format of the consensus meeting. In particular, our results suggest that face-to-face consensus meetings do not guarantee a better consensus, at least if one considers an outcome where the opinions of all reviewers involved would weigh equally in the final score (*Gallo et al., 2013*). Our results also suggest that the level of (dis)agreement among reviewers is more dependent on the type of action or the scientific panel, rather than the way peer review is organized.

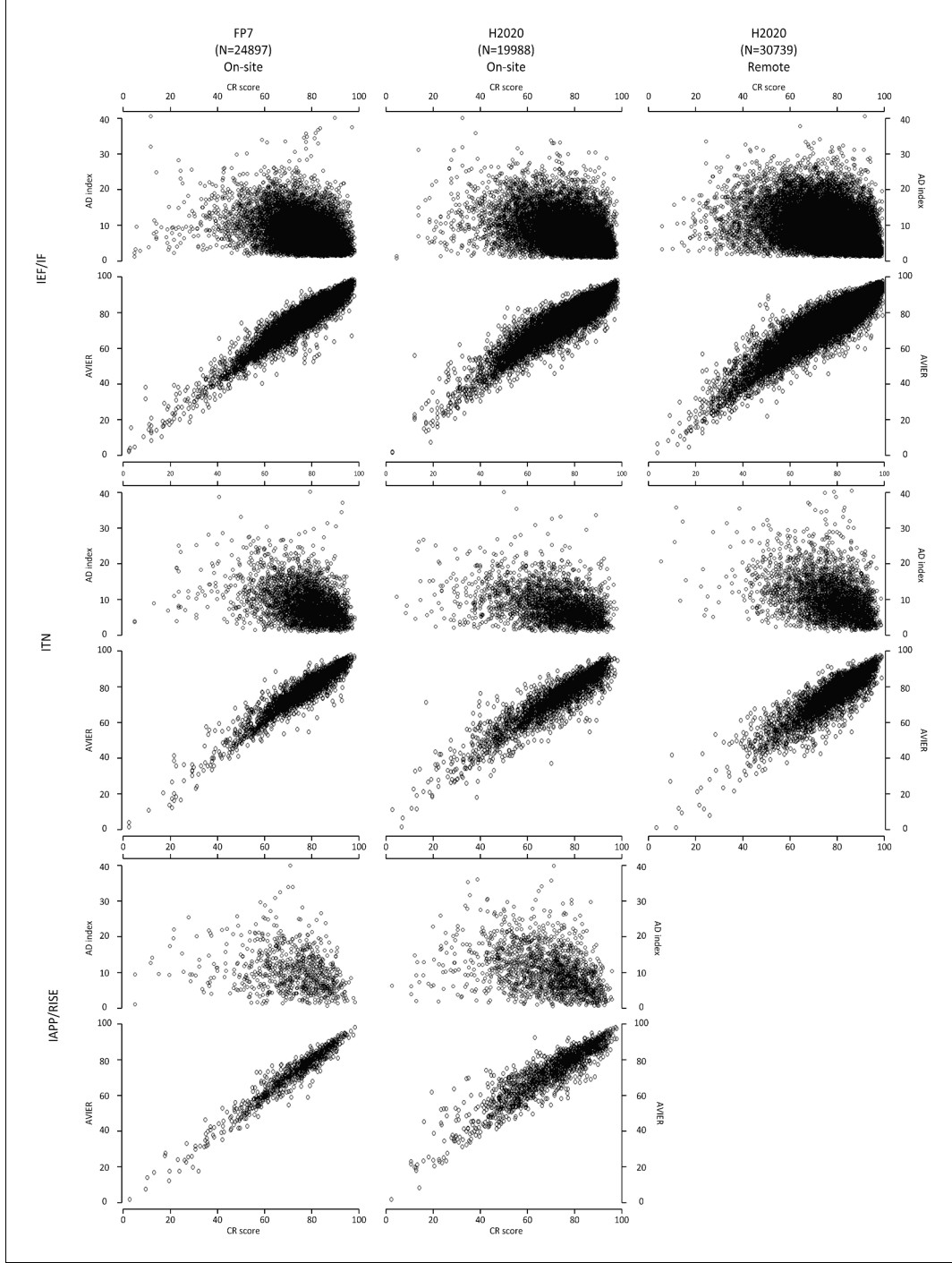

**Figure 3.** Scatter plots showing AD indices and the average of IER scores versus CR scores for the three different Marie Curie actions between 2007 and 2018. The upper panel plots average deviation (AD) indices versus Consensus Report (CR) scores; the lower panel plots the average of the Individual Evaluation Reports (AVIER) scores versus the CR scores. The distributions are shown for the three Marie Curie actions (horizontally) during different time periods (vertically). The Results section describes how these actions changed over this period.

Our study has some limitations. As a retrospective analysis focusing only on reviewer scores, it cannot provide an insight into the reasons why reviewers agree or disagree on a particular proposal. Reviewers may have diverse perceptions of their role during the evaluation

**Table 4.** CR scores (mean and SD), broken down by level of agreement between reviewers, for the three different H2020 Marie Skłodowska-Curie actions between 2014 and 2018.

| | | Mean CR score (SD), number (%) | | |
|---|---|---|---|---|
| | | **Full agreement** | **No agreement** | **Other** |
| 2014 | IF (n = 7,397) | 85.2 (10.0), n = 2,032 (27.5%) | 69.7 (12.9), n = 771 (10.4%) | 79.1 (11.8), n = 4,594 (62.1%) |
| | ITN (n = 1,149) | 84.1 (9.4), n = 309 (26.9%) | 71.8 (11.3), n = 136 (11.8%) | 78.6 (11.5), n = 704 (61.3%) |
| | RISE (n = 200) | 83.3 (12.6), n = 23 (11.5%) | 65.5 (13.7), n = 51 (25.5%) | 70.1 (12.5), n = 126 (63.0%) |
| 2015 | IF (n = 8,364) | 84.8 (10.6), n = 2,242 (26.8%) | 70.9 (13.7), n = 950 (11.6%) | 78.8 (12.5), n = 5,172 (61.8%) |
| | ITN (n = 1,558) | 84.1 (10.0), n = 413 (26.5%) | 74.1 (12.2), n = 159 (10.2%) | 79.7 (11.3), n = 986 (63.3%) |
| | RISE (n = 361) | 81.1 (15.0), n = 66 (18.3%) | 68.4 (12.4), n = 64 (17.7%) | 73.3 (15.1), n = 231 (64.0%) |
| 2016 | IF (n = 8,805) | 85.4 (10.7), n = 2,189 (24.9%) | 71.0 (13.4), n = 1,117 (12.7%) | 79.0 (12.3), n = 5,499 (62.4%) |
| | RISE (n = 366) | 79.1 (15.0), n = 73 (19.9%) | 67.3 (13.9), n = 68 (18.6%) | 74.8 (14.6), n = 225 (61.5%) |
| 2017 | IF (n = 8,940) | 85.1 (11.0), n = 2,142 (23.9%) | 69.6 (13.5), n = 1,125 (12.6%) | 78.2 (12.9), n = 5,673 (63.5%) |
| | ITN (n = 1,702) | 86.7 (8.5), n = 473 (27.8%) | 72.7 (11.6), n = 141 (8.3%) | 81.2 (10.4), n = 1,088 (63.9%) |
| | RISE (n = 321) | 82.4 (10.5), n = 61 (19.0%) | 64.1 (14.7), n = 57 (17.8%) | 73.5 (14.4), n = 203 (63.2%) |
| 2018 | IF (n = 9,658) | 85.2 (11.6), n = 2,345 (24.3%) | 69.3 (13.4), n = 1,247 (12.9%) | 78.2 (13.4), n = 6,066 (62.8%) |
| | ITN (n = 1,634) | 86.5 (9.1), n = 427 (26.1%) | 72.7 (11.0), n = 161 (9.9%) | 81.2 (10.1), n = 1,046 (64.0%) |
| | RISE (n = 272) | 80.4 (16.8), n = 45 (16.5%) | 65.2 (14.1), n = 37 (13.6%) | 73.2 (14.0), n = 190 (69.9%) |

* One-way ANOVA, all differences significant at p<0.001 level. The method to divide proposals between Full Agreement, No Agreement and Other is described in the Results section.

CR: Consensus Report; SD: standard deviation.

process, and/or interpret the evaluation criteria differently (*Abdoul et al., 2012*). Disagreement could also arise from inherent characteristics of the proposals, with a recent study showing that interdisciplinary proposals tend to score lower (*Bromham et al., 2016*), or from reviewers taking a conservative approach to controversial proposals (*Luukkonen, 2012*). Also, our analysis does not explore if proposals with higher scores are more likely to be successful (in terms of future outputs) than proposals with lower scores. Indeed, the ability of peer review to predict future success, as measured by scientific productivity and impact, has been subject to contradictory findings (*Bornmann and Daniel, 2005*; *Bornmann et al., 2008*; *Li and Agha, 2015*; *Fang et al., 2016*; *Lindner and Nakamura, 2015*; *van den Besselaar and Sandström, 2015*). Finally, projects funded by the various Marie Curie actions require researchers to be mobile, and this might limit the relevance of our findings to other grant evaluation systems, though we have tried to guard against this by analysing three different types of actions, each with different levels of complexity and different success rates.

The MSCA evaluation process is currently evolving towards a system in which reviewers write Individual Evaluation Reports that do not contain numerical scores. The IF action started operating this way in 2019, and ITN and RISE followed in 2020. Although it will no longer be possible to undertake the sort of retrospective analysis of IER and CR scores we have performed here, as IER will not have scores anymore, it will still be possible to observe the overall distribution of CR scores over time and thus monitor the consistency of the process comparing current/future evaluation exercises with previous ones. We also suggest performing such analyses for other EU funding research programmes, as happens at other major funding agencies (*Pier et al., 2018*; *Kaplan et al., 2008*; *Fang et al., 2016*; *Lindner and Nakamura, 2015*; *Martin et al., 2010*). This would improve our understanding of the use of peer review to evaluate grant applications and proposals (*Azoulay, 2012*). The COVID-19 pandemic means that the use of remote consensus meetings is likely to increase under Horizon Europe, the successor to H2020 (*European Commission, 2020*). As such, this study gives us confidence that the outcomes of the grant peer review process will not be impacted by this change.

**Table 5.** CR scores (mean and SD), broken down by level of agreement between reviewers and scientific panel, for the three different H2020 Marie Skłodowska-Curie actions for 2014–2015 and 2016–2018.

| | | Mean CR score (SD) | | | | | |
| --- | --- | --- | --- | --- | --- | --- | --- |
| | | **Full agreement** | | **No agreement** | | **Other** | |
| **IF** | | **2014–15** | **2016–18** | **2014–15** | **2016–18** | **2014–15** | **2016–18** |
| | Overall | 85.0 (10.4) | 85.2 (11.1) | 70.4 (13.3) | 70.0 (13.4) | 78.9 (12.2) | 78.5 (12.9) |
| | CHE | 84.4 (9.4) | 85.2 (10.5) | 70.9 (13.0) | 70.5 (13.8) | 78.8 (10.9) | 79.1 (12.1) |
| | ECOSOC | 85.2 (12.3) | 84.0 (13.6) | 70.0 (14.2) | 69.7 (13.9) | 78.1 (13.7) | 76.3 (14.7) |
| | ENG | 83.4 (11.8) | 83.6 (12.8) | 67.8 (12.7) | 68.2 (13.2) | 76.1 (13.3) | 77.1 (14.2) |
| | ENV | 85.0 (10.7) | 85.4 (10.8) | 70.3 (13.7) | 70.2 (13.2) | 78.7 (12.3) | 79.5 (12.4) |
| | LIF | 85.7 (9.1) | 86.7 (10.2) | 72.1 (12.5) | 71.4 (12.9) | 80.4 (11.0) | 80.1 (11.5) |
| | MAT | 83.1 (12.3) | 85.5 (9.9) | 70.1 (14.6) | 68.9 (14.1) | 79.5 (12.6) | 78.9 (12.1) |
| | PHY | 84.8 (9.5) | 85.2 (8.4) | 72.5 (11.3) | 69.1 (12.3) | 80.4 (10.0) | 78.8 (10.7) |
| **ITN** | | **2014–15** | **2017–18** | **2014–15** | **2017–18** | **2014–15** | **2017–18** |
| | Overall | 84.1 (9.8) | 86.6 (8.8) | 73.0 (11.8) | 73.8 (11.3) | 79.2 (11.4) | 81.4 (10.3) |
| | CHE | 84.6 (7.7) | 87.8 (8.0) | 73.6 (13.0) | 76.9 (10.2) | 80.8 (10.0) | 84.3 (8.3) |
| | ECOSOC | 85.2 (9.7) | 85.1 (11.8) | 73.5 (13.4) | 73.7 (11.5) | 77.1 (13.7) | 80.1 (12.3) |
| | ENG | 83.1 (11.5) | 84.3 (9.6) | 71.0 (12.3) | 71.9 (11.8) | 77.9 (11.9) | 80.1 (10.0) |
| | ENV | 84.3 (8.5) | 89.1 (7.4) | 71.3 (12.8) | 71.9 (11.8) | 79.3 (10.8) | 82.2 (10.5) |
| | LIF | 84.0 (10.2) | 87.8 (7.5) | 74.3 (10.0) | 72.7 (13.0) | 80.4 (11.0) | 81.6 (10.2) |
| | MAT | 79.3 (10.4) | 85.6 (8.8) | 76.2 (6.3) | 70.9 (6.0) | 76.5 (9.9) | 77.3 (10.6) |
| | PHY | 85.7 (8.3) | 87.1 (7.2) | 77.1 (9.5) | 74.1 (11.7) | 81.2 (9.6) | 82.9 (9.0) |
| **RISE** | | **2014–18** | | **2014–18** | | **2014–18** | |
| | Overall | 80.9 (14.2) | | 66.3 (13.9) | | 73.3 (14.9) | |
| | CHE | 82.6 (11.3) | | 67.3 (16.3) | | 77.9 (12.4) | |
| | ECOSOC | 81.1 (13.5) | | 65.5 (13.7) | | 70.5 (16.4) | |
| | ENG | 78.0 (15.8) | | 66.0 (14.7) | | 72.3 (15.6) | |
| | ENV | 80.2 (19.1) | | 67.5 (11.2) | | 74.8 (14.4) | |
| | LIF | 79.9 (11.1) | | 68.4 (13.8) | | 72.4 (14.8) | |
| | MAT | 84.4 (4.9) | | 57.6 (14.2) | | 74.4 (12.0) | |
| | PHY | 65.9 (13.5) | | 65.9 (13.5) | | 76.8 (10.5) | |

* One-way ANOVA, all differences were significant at p<0.001 level. CR: Consensus Report; SD: standard deviation. CHE: Chemistry; ECOSOC: Economics and social sciences; ENG: Engineering; ENV: Environment; LIF: Life sciences; MAT: Mathematics; PHY: Physics.

## Methods

### The EU's Marie Curie funding programme

The data for this study consisted of 24,897 proposals evaluated under MCA (IEF, ITN, IAPP; *Pina et al., 2015*) as part of the Seventh Framework Programme (FP7; 2007–2013), and 50,727 proposals evaluated under MSCA (IF, ITN, RISE; *Table 1*) as part of Horizon 2020 (H2020; 2014–2020). The Intra-European Fellowships (IEF) action and the Individual Fellowships (IF) action funded individual postdoctoral fellowships for mobile researchers. The Initial Training Networks (ITN) action and the Innovative Training Networks (also ITN) action funded projects that trained mobile doctoral candidates. The Industry-Academia Pathways and Partnerships (IAPP) action and the Research and Innovation Staff Exchange (RISE) action funded projects that promoted the mobility of staff between organizations from both public and private sectors.

The scoring scale used in FP7 and H2020 was based on five ordinal qualitative descriptors (0=fail, 1=poor, 2=fair, 3=good, 4=very good, and 5=excellent), with reviewers scoring MCA/MSCA proposals with one-digit decimal. In that context, a difference of 0.5 points or less (ie, 10 points or less when converted to a 0–100 scale

for the final IER and CR scores) can be considered as a reasonably good agreement. The evaluation criteria used under FP7 were: (i) scientific and technological quality; (ii) training (ITN and IEF) or transfer of knowledge (IAPP); (iii) implementation; (iv) impact; (v) fellow candidate's CV (IEF only). The evaluation criteria used under H2020: (i) excellence; (ii) impact; (iii) implementation.

MCA/MSCA proposals were evaluated within one of the following panels: chemistry (CHE), economic sciences (ECO), information science and engineering (ENG), environment and geo-sciences (ENV), life sciences (LIF), mathematics (MAT), physics (PHY), and social sciences and humanities (SOC). For most of the period analysed, proposals in economic sciences and in social sciences and humanities were evaluated by the same pool of reviewers, so we have treated ECO and SOC as a single panel for the purposes of this study.

### Data and analyses

The dataset in *Supplementary file 1* includes data on all the proposals (n = 75,624) analysed in this study, sorted by type of action, call year and scientific panel. For each proposal, scores for the Consensus Report (CR) and the respective scores given by reviewers in their Individual Evaluation Report (IER) are reported. All analyses were performed with JASP statistical software v. 0.11.1.0. (*JASP team, 2020*), R v.3.6.3. (*R Development Core Team, 2020*), and SPSS Statistics for Windows v.19.0 (*Corp, 2010*).

### Disclaimer

All views expressed in this article are strictly those of the authors and may in no circumstances be regarded as an official position of the Research Executive Agency or the European Commission.

**David G Pina** is in the Research Executive Agency, European Commission, Brussels, Belgium

David.Pina@ec.europa.eu

https://orcid.org/0000-0002-4930-748X

**Ivan Buljan** is in the Department for Research in Biomedicine and Health, University of Split School of Medicine, Split, Croatia

https://orcid.org/0000-0002-8719-7277

**Darko Hren** is in the Department of Psychology, University of Split School of Humanities and Social Sciences, Split, Croatia

https://orcid.org/0000-0001-6465-6568

**Ana Marušić** is in the Department for Research in Biomedicine and Health, University of Split School of Medicine, Split, Croatia

https://orcid.org/0000-0001-6272-0917

*Author contributions:* David G Pina, Conceptualization, Data curation, Formal analysis, Investigation, Methodology, Writing - original draft, Writing - review and editing; Ivan Buljan, Conceptualization, Data curation, Formal analysis, Investigation, Visualization, Methodology, Writing - original draft, Writing - review and editing; Darko Hren, Formal analysis, Visualization, Methodology, Writing - review and editing; Ana Marušić, Conceptualization, Formal analysis, Funding acquisition, Investigation, Methodology, Writing - original draft, Writing - review and editing

*Competing interests:* The authors declare that no competing interests exist.

### Funding

| Funder | Grant reference number | Author |
|---|---|---|
| Hrvatska Zaklada za Znanost | IP-2019-04-4882 | Ana Marušić |

The funders had no role in study design, data collection and interpretation, or the decision to submit the work for publication.

**Decision letter and Author response**
Decision letter https://doi.org/10.7554/eLife.59338.sa1
Author response https://doi.org/10.7554/eLife.59338.sa2

## Additional files

### Supplementary files

• Supplementary file 1. Evaluation scores for all the Marie Curie proposals (n = 75,624) analysed in this study. For each proposal, the total score for the Consensus Report (CR) and the respective scores given by reviewers in their Individual Evaluation Report (IER) are reported.

• Supplementary file 2. Table A1 and Table A2.

• Transparent reporting form

### Data availability

All data presented in this study are included in the manuscript and supporting files.

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
