## [Decision Letter]

Thank you for submitting your article "Meta-research: On-site or remote grant evaluation do not affect reviewers' agreement and scores of Marie Curie proposals" to *eLife* for consideration as a Feature Article. Your article has been reviewed by three peer reviewers, and the evaluation has been overseen by Cassidy Sugimoto as the Reviewing Editor. The following individual involved in review of your submission has agreed to reveal their identity: Jesper Wiborg Schneider (Reviewer #3).

The reviewers have discussed the reviews with each other and the Reviewing Editor has drafted this decision letter to help you prepare a revised submission.

Summary:

This paper aims at analyzing the effect of location of review (on-site or remote) on reviewers' agreements and evaluation, using proposals submitted to the Marie (Skłodowska-)Curie Actions proposals under the EU's Seventh Research Framework program (FP7; 2007-2013) and then the Horizon 2020 (H2020; 2014-2018) framework program for research and innovation over a period of 12 year. The analysis is performed using time-series analysis, comparing the Seventh Framework Programme (25,000 proposals) and Horizon 2020 (52,000 proposals), focusing on the difference between onsite and remote meetings. Results show that no important differences were observed between the organization of reviews, and that reviewers' agreement were similar in all cases.

The creation of this evidence-base for directing policy decisions is to be highly commended. And importantly, the variables examined were very stable over the observed period, providing some evidence that the move from the FP7 program to the H2020 was not greatly affected over time. In the current COVID situation, these findings have clear policy implications for the grant review process. Significant revisions, however, are necessary to improve the quality of the manuscript and the robustness of the results.

Essential revisions:

1) The work lacks proper motivation and contextualization with the current literature. Several highly relevant contemporary studies have been overlooked as well as fundamental research within the area. Some suggested references include:

Witteman, H. O., Hendricks, M., Straus, S., and Tannenbaum, C. (2019). Are gender gaps due to evaluations of the applicant or the science? A natural experiment at a national funding agency. The Lancet, 393(10171), 531-540.

Lee, C. J., Sugimoto, C. R., Zhang, G., and Cronin, B. (2013). Bias in peer review. Journal of the American Society for Information Science and Technology, 64(1), 2-17.

Murray, D., Siler, K., Larivière, V., Chan, W. M., Collings, A. M., Raymond, J., and Sugimoto, C. R. (2019). Gender and international diversity improves equity in peer review. BioRxiv, 400515.

The work generally motivates on the assumption that there is "fear that shifting from on-site to remote consensus meetings negatively affects the quality and the robustness of the evaluation process." This motivation could be strengthened with engagement with the literature, particularly that which reviews mode shifting as well as different approaches to peer review. While the rationale for the study is implied, it should be made explicit.

2) *eLife* is read by an international audience. Therefore, more information on the evaluation process is warranted, to ensure that the readership is familiar with the various programs and systems. A schematic representation or references to external sources would be helpful. The use of acronyms and the changing variables over time make it difficult to understand and evaluate.

3) A limitation of the present analysis is on the particularistic aspects of the program (i.e., an elite program focused on young researchers). The authors should discuss generalizability and potential limitations. The broader meta-research interests of this study should be emphasized.

4) The main focus of this study is on the comparison between remote and in-person reviewing. However, there seems to be considerable ambiguity in the comparison. Questions regarding the data are listed below:

a) Data from the FP7 program is only from in person reviews, whereas data from the 2014-2016 H2020 program includes in-person and remote reviews. It is not entirely clear what the proportion of in person and remote reviews was. Then, it seems there is a complete shift to remote reviews happened in 2016 for H2020. While initially some comparisons are listed in the manuscript between the H2020 program before and after the 2016 switch, later comparisons seem to be made between all FP7 data and all H2020 data (Table 1 and Figure 2), which presumably are not fair comparisons (at least to address the effect of review format), because some of the H2020 data are in person and some are remote. Particularly, it would be useful to include histogram comparisons of before 2016 and after for H2020.

b) In addition, it seems in 2016, for the ITN program 4 reviewers were used instead of 3 for other H2020 years, as well as 3 for FP7. Yet it still seems the ITN data with 4 reviewers was included in the analysis for Figure 2, but was removed for the agreement analysis in Table 2/3. It is unclear why this data wasn't removed from the analysis in Figure 2, as it is unclear what effect different # of reviewers might have on these histograms.

c) Other differences exist as well, including 4 criteria used for FP7 and 3 for H2020, while individual fellowships had a fifth criterion. Also, different Actions had different weighting of criteria scores to calculate the overall score, and therefore it is not entirely clear these comparisons are appropriate.

d) Also, the number of proposals (and presumably the funding success rates) are significantly different between FP7 and H2020, but how this effects the results is not discussed.

e) In Figure 3, again it seems all the H2020 data are compared to FP7. Also, it looks as though the mean square error for H2020 in Figure 3 is larger than FP7, perhaps indicating more robust/impactful discussion. It may be interesting to compare these results to other work examining scoring patterns before and after discussion for in person and remote panels (Carpenter et al., 2015).

f) The use of different reviewer scores are generally hard to follow. It is not clear when the authors analyze the proposals based on reviewer scores from the Individual Evaluation Reports (IER) or the Consensus Report (CR). One example are the analyses based on CR scores and AD indices presented in Table 1. The authors show that the mean CR score appears stable between the two periods, but also that the Absolute Deviation (AD) Index is stable. The CR score is obviously the consensus score from the reviewers' consensus meeting, but the AD index must be from the Individual Evaluation Reports (?). Why would we expect these individual review scores, and the deviation between them, to differ across the time periods when the policy change relates to the way consensus meetings are conducted? This is important as the authors argue that consensus meetings could feasibly be held remotely because review score deviations does not depend on a remote/non-remote meeting. This comes across as largely a strawman argument.

5) There are also several concerns about analysis:

a) As there are many changing factors, a bit more focus on directly comparable data sub-sets (controlling as many variables as possible) which are centered on a specific issue (in-person vs remote reviews, 4 vs 3 reviewers, 4 vs 3 criteria, hi/lo success rate etc.) would be helpful. It may be useful to use a multi-level regression approach to examine blocks of factors and their relative effect on CR score, AVIER-CR score and AD so one can more easily assess their importance.

b) The authors argue that proposals with higher CR scores also exhibit lower reviewer disagreement (i.e. low AD index). Figure 3 shows the raw scatter plot of these variables, but it seems that the correlation is very week, with e.g. AD scores of 10 being widely scattered across CR values of 50-100.

c) In the Transparent reporting form the authors state that they "did not perform sample size estimation" because they included "[a]ll available proposals from 2007 to 2018" in the analysis and therefore analyses the "entire population". Nevertheless, frequentists inferential statistics are employed which seem odd; there is no real stochastic data generation mechanism to be studied. It is not clear how CI and p-values should be interpreted in this context – it would be some imaginary sampling process to a non-existing super-population which makes it all a bit far out for a proper comprehension.

d) It is not clear how the comparison strategy for interrupted-time-series were planned. One could argue that the lumping together of all years before and after dampens any effect if indeed such an effect were present.

6) What evidence is available regarding the construction of aggregate consensus scores? Can the authors discuss why "consensus negotiations" seem to end up in the same region repeatedly?

7) Finally, the authors also do not discuss the likelihood that subtle differences between remote and in person reviews might still exist that are not detected with these analyses. For instance, it has been reported that discussion time is shorter for remote reviews (Gallo et al., 2013), which may suggest less robust reviews. Less discussion may also yield more biased reviews, which would represent a decrease in review quality.

8) While it seems rather obvious for the reader that there is no practical difference between conditions in most of the comparisons, yet the authors seem to make the claims based on the "statistical significance filter". That is problematic for several reasons; one is given above in point 7. Another is the logical fallacy of claiming "no difference" due to a failure to reject the null hypothesis. Yes, the difference in minuscule -0,.22 [CI = -1.47 – 1.02] – but failing to reject a null hypothesis cannot be translated into a claim of "no difference" – that claim has to be warranted by other means. Interestingly, just below this statement is one that firmly claims an effect, this time the difference is 1.06 points [CI = 0.75 – 1.37] and it is "statistically significant". This is selection on "significance". Yes, the point estimate is more precisely estimated but it is still very low. Which raise the question of why the "non-significant" estimate above fails the test given it large sample size? Why study the compatibility of data under (a false) H0 in an imaginary infinite random sampling procedure?

9) This also undermines the statements where the authors claim that the large sample sizes produce the "statistical significance" (which in principle is correct), and then further state that "so the statistical differences can be attributed to sample size rather than actual meaningful differences". "Statistical significance" is a complicated and misunderstood concept. It is very hard to interpret when examining non-experimental data such as an apparent population. And "statistical differences" – no matter how they are produced – cannot tell you to what extent such differences might be meaningful.

10) To enhance the policy relevance, the work should engage more deeply with the contemporary literature in the discussion. The work may also want to particularly engage with literature that discusses mode shifting in the pandemic across several sectors. This will help situate the contribution more firmly.

---

## [Author Response]

Essential revisions:1) The work lacks proper motivation and contextualization with the current literature. Several highly relevant contemporary studies have been overlooked as well as fundamental research within the area. Some suggested references include:Witteman, H. O., Hendricks, M., Straus, S., and Tannenbaum, C. (2019). Are gender gaps due to evaluations of the applicant or the science? A natural experiment at a national funding agency. The Lancet, 393(10171), 531-540.Lee, C. J., Sugimoto, C. R., Zhang, G., and Cronin, B. (2013). Bias in peer review. Journal of the American Society for Information Science and Technology, 64(1), 2-17.Murray, D., Siler, K., Larivière, V., Chan, W. M., Collings, A. M., Raymond, J., and Sugimoto, C. R. (2019). Gender and international diversity improves equity in peer review. BioRxiv, 400515.The work generally motivates on the assumption that there is "fear that shifting from on-site to remote consensus meetings negatively affects the quality and the robustness of the evaluation process." This motivation could be strengthened with engagement with the literature, particularly that which reviews mode shifting as well as different approaches to peer review. While the rationale for the study is implied, it should be made explicit.

We thank the editors and reviewers for their observation, and we redrafted the Introduction in order to focus on the main purpose of the study, which is to assess whether organizational changes in the evaluation process affected the scoring behaviour and level of agreement among reviewers. The comment that there is “fear that shifting {…] evaluation process” has been removed.

2) eLife is read by an international audience. Therefore, more information on the evaluation process is warranted, to ensure that the readership is familiar with the various programs and systems. A schematic representation or references to external sources would be helpful. The use of acronyms and the changing variables over time make it difficult to understand and evaluate.

We thank the editors and reviewers for suggesting to add a schematic representation of the data analysed. We now introduced a new Table 1, where we indicated, for each year of evaluation (call year) and type of actions under study, the number of proposals, number of evaluation criteria applicable for that call, and the review format (on-site or remote consensus). We also describe MSCA grant system in detail in the Analytical Framework part of the Methods section.

3) A limitation of the present analysis is on the particularistic aspects of the program (i.e., an elite program focused on young researchers). The authors should discuss generalizability and potential limitations. The broader meta-research interests of this study should be emphasized.

We thank the editors and reviewers for flagging this study limitation. We have addressed this in the Discussion section where we state that the results cannot be generalized to other grant systems. Nevertheless, we would like to stress the fact that the MSCA program has an important global impact on research, as mobility funding is not restricted to Europe but to the global scientific community.

4) The main focus of this study is on the comparison between remote and in-person reviewing. However, there seems to be considerable ambiguity in the comparison. Questions regarding the data are listed below:a) Data from the FP7 program is only from in person reviews, whereas data from the 2014-2016 H2020 program includes in-person and remote reviews. It is not entirely clear what the proportion of in person and remote reviews was. Then, it seems there is a complete shift to remote reviews happened in 2016 for H2020. While initially some comparisons are listed in the manuscript between the H2020 program before and after the 2016 switch, later comparisons seem to be made between all FP7 data and all H2020 data (Table 1 and Figure 2), which presumably are not fair comparisons (at least to address the effect of review format), because some of the H2020 data are in person and some are remote. Particularly, it would be useful to include histogram comparisons of before 2016 and after for H2020.

We thank the editors and reviewers for this observation and acknowledge the need to disambiguate the data under study. We now reorganized the study, with each type of action being analysed separately, throughout the article, and by focusing on the effect of the two relevant organizational changes occurring over time, i.e. the reduction of the number of evaluation criteria in 2014, and the shift in the consensus format, from on-site to remote consensus meetings, for ITN and IF (happening in 2016-17). All figures and tables are now presented with the data of the three type of actions shown separately. We also changed the title of the manuscript to reflect these changes.

b) In addition, it seems in 2016, for the ITN program 4 reviewers were used instead of 3 for other H2020 years, as well as 3 for FP7. Yet it still seems the ITN data with 4 reviewers was included in the analysis for Figure 2, but was removed for the agreement analysis in Table 2/3. It is unclear why this data wasn't removed from the analysis in Figure 2, as it is unclear what effect different # of reviewers might have on these histograms.

We thank the editors and reviewers for this remark. In order to make the analysis coherent and comparable, we excluded the data related to ITN 2016 from all analyses.

c) Other differences exist as well, including 4 criteria used for FP7 and 3 for H2020, while individual fellowships had a fifth criterion. Also, different Actions had different weighting of criteria scores to calculate the overall score, and therefore it is not entirely clear these comparisons are appropriate.

The difference in the number of evaluation criteria is now one of the two organizational changes under study, together with the transition from on-site to remote consensus for ITN and IF. Also, as mentioned under point 4.a. the three type of actions ( IEF/IF, ITN and IAPP/RISE) are now presented separately, which makes the analysis of the effect of the reduction of the number of evaluation criteria possible. Therefore, the analysis allows now for the assessment of a single change at the time, for each of the three type of actions.

d) Also, the number of proposals (and presumably the funding success rates) are significantly different between FP7 and H2020, but how this effects the results is not discussed.

Success rates are not so different between FP7 and H2020. The number of proposals generally increased over the years, but so did the available funding. The differences in success rates are mostly a characteristic of the type of action. For instance, ITN remained always the most competitive with a success rate around 10%, both in FP7 and H2020. We now clarify this in more detail in the Analytical framework paragraph under Methods section.

e) In Figure 3, again it seems all the H2020 data are compared to FP7. Also, it looks as though the mean square error for H2020 in Figure 3 is larger than FP7, perhaps indicating more robust/impactful discussion. It may be interesting to compare these results to other work examining scoring patterns before and after discussion for in person and remote panels (Carpenter et al., 2015).

We thank the editors and reviewers for this remark. As reported for point 4)a, we now decided to present the results separately for each of the three type of actions (the new Figure 3). Therefore, the data are not presented anymore as aggregate numbers for FP7 vs H2020, but rather as data for IEF/IF, ITN and IAPP/RISE. We also assessed the two organizational changes observed over time for each of the action type.

f) The use of different reviewer scores are generally hard to follow. It is not clear when the authors analyze the proposals based on reviewer scores from the Individual Evaluation Reports (IER) or the Consensus Report (CR). One example are the analyses based on CR scores and AD indices presented in Table 1. The authors show that the mean CR score appears stable between the two periods, but also that the Absolute Deviation (AD) Index is stable. The CR score is obviously the consensus score from the reviewers' consensus meeting, but the AD index must be from the Individual Evaluation Reports (?). Why would we expect these individual review scores, and the deviation between them, to differ across the time periods when the policy change relates to the way consensus meetings are conducted? This is important as the authors argue that consensus meetings could feasibly be held remotely because review score deviations does not depend on a remote/non-remote meeting. This comes across as largely a strawman argument.

Thank you for your comment. As a result of the way how results are presented in this new version of the manuscript, by type of action and considering the two changing events in the evaluation process, all tables and figures were reworked accordingly.

We agree with the reviewers that the AD index is related to the IER phase and as such should not be affected by the way consensus meetings are organized. However, we also assessed the effect of the transition from FP7 to H2020 where the number of criteria changed and that may have had an effect on individual raters' agreement (i.e. AD indices). Furthermore, we drew our conclusions about the effect of change in the organization of consensus meeting from the CR scores (Table 2) and the differences between CR scores and average IER scores (Figure 2A). In that sense, the AD indices show evidence about the stability of the IER phase. This is an important piece of information for building our argument. In case that we had found the effect, the stable AD indices would strengthen the argument for attributing that effect to the change from the "on-site" to "remote" consensus meeting. Given that we found no effect, the AD indices seem superfluous, yet we could not make any conclusions if we did not have evidence of basic stability of IER phase.

5) There are also several concerns about analysis:a) As there are many changing factors, a bit more focus on directly comparable data sub-sets (controlling as many variables as possible) which are centered on a specific issue (in-person vs remote reviews, 4 vs 3 reviewers, 4 vs 3 criteria, hi/lo success rate etc.) would be helpful. It may be useful to use a multi-level regression approach to examine blocks of factors and their relative effect on CR score, AVIER-CR score and AD so one can more easily assess their importance.

Thank you for your comment. We agree that a multilevel analysis would be very interesting; however, we do not have the proper data to perform a multilevel regression analysis because if we would like to have a comparison between the high and low success rates, 3 vs 4 reviewers and across the calls, we should have all combinations for each project call. For example, we should have IAPP/RISE which would have low success rate and 3 reviewers vs IAPP/RISE with high success and 3 reviewers, then the same combination with 4 reviewers, and the same for IF and ITN.

As it can be noted from Table 1, that condition was not satisfied and therefore we decided to provide more granular data for each call by analysing them separately. So, presenting the results across the calls is already the segmentation of scores across smaller parts, and we used the same procedure to examine the differences across different calls.

b) The authors argue that proposals with higher CR scores also exhibit lower reviewer disagreement (i.e. low AD index). Figure 3 shows the raw scatter plot of these variables, but it seems that the correlation is very week, with e.g. AD scores of 10 being widely scattered across CR values of 50-100.

Thank you for your comment. This statement was removed in the revised manuscript.

c) In the "transparent reporting form" the authors state that they "did not perform sample size estimation" because they included "[a]ll available proposals from 2007 to 2018" in the analysis and therefore analyses the "entire population". Nevertheless, frequentists inferential statistics are employed which seem odd; there is no real stochastic data generation mechanism to be studied. It is not clear how CI and p-values should be interpreted in this context – it would be some imaginary sampling process to a non-existing super-population which makes it all a bit far out for a proper comprehension.

Confidence intervals were removed, and we introduced standard deviation as a measure of dispersion, as we analysed the whole dataset (the whole population). However, in the ITS analysis we decided to keep the confidence intervals to assess the significance of the differences, as well as the potential size of the differences.

d) It is not clear how the comparison strategy for interrupted-time-series were planned. One could argue that the lumping together of all years before and after dampens any effect if indeed such an effect were present.

Thank you for your comment. We refined our analysis across the calls now, and we provided the additional explanation for ITS procedure. Now we have separate comparisons for separate calls and this provides a comprehensive overview of the assessment shift across calls.

6) What evidence is available regarding the construction of aggregate consensus scores? Can the authors discuss why "consensus negotiations" seem to end up in the same region repeatedly?

Thank you for your comment. This is now resolved by separate ITS analysis and the answer to this question is under 5)d.

7) Finally, the authors also do not discuss the likelihood that subtle differences between remote and in person reviews might still exist that are not detected with these analyses. For instance, it has been reported that discussion time is shorter for remote reviews (Gallo et al., 2013), which may suggest less robust reviews. Less discussion may also yield more biased reviews, which would represent a decrease in review quality.

We thank the editors and reviewers for this pertinent observation. In the context of the Marie Curie programme, the remote consensus meetings are organized in a way that they have the same discussion time as when the meetings were held on-site, face-to-face. A statement on this issue has been added to the Discussion section of the revised manuscript.

8) While it seems rather obvious for the reader that there is no practical difference between conditions in most of the comparisons, yet the authors seem to make the claims based on the "statistical significance filter". That is problematic for several reasons; one is given above in point 7. Another is the logical fallacy of claiming "no difference" due to a failure to reject the null hypothesis. Yes, the difference in minuscule -0,.22 [CI = -1.47 – 1.02] – but failing to reject a null hypothesis cannot be translated into a claim of "no difference" – that claim has to be warranted by other means. Interestingly, just below this statement is one that firmly claims an effect, this time the difference is 1.06 points [CI = 0.75 – 1.37] and it is "statistically significant". This is selection on "significance". Yes, the point estimate is more precisely estimated but it is still very low. Which raise the question of why the "non-significant" estimate above fails the test given it large sample size? Why study the compatibility of data under (a false) H0 in an imaginary infinite random sampling procedure?

We thank the editors and reviewers for this remark. We redrafted the text to state that the differences were significant but extremely small. Furthermore, instead of stating that there was no difference between the groups, we now state that the scores were similar between groups.

9) This also undermines the statements where the authors claim that the large sample sizes produce the "statistical significance" (which in principle is correct), and then further state that "so the statistical differences can be attributed to sample size rather than actual meaningful differences". "Statistical significance" is a complicated and misunderstood concept. It is very hard to interpret when examining non-experimental data such as an apparent population. And "statistical differences" – no matter how they are produced – cannot tell you to what extent such differences might be meaningful.

We agree with the editors and reviewers about the difficulty in interpreting the concept of “statistical significance”. Therefore, we decided to remove this phrase in the text so that it does not cause confusion.

10) To enhance the policy relevance, the work should engage more deeply with the contemporary literature in the discussion. The work may also want to particularly engage with literature that discusses mode shifting in the pandemic across several sectors. This will help situate the contribution more firmly.

We thank the editors and reviewers in proposing to expand the discussion in view of the current COVID-19 pandemic. We highlighted relevant implication that this study entails in the revised manuscript.